# Perceived Change in Tobacco Use and Its Associated Factors among Older Adults Residing in Rohingya Refugee Camps during the COVID-19 Pandemic in Bangladesh

**DOI:** 10.3390/ijerph182312349

**Published:** 2021-11-24

**Authors:** Sabuj Kanti Mistry, ARM Mehrab Ali, Uday Narayan Yadav, Md. Nazmul Huda, Saruna Ghimire, Md. Ashfikur Rahman, Sompa Reza, Rumana Huque, Muhammad Aziz Rahman

**Affiliations:** 1ARCED Foundation, Dhaka 1216, Bangladesh; mehrabbabu@gmail.com; 2Centre for Primary Health Care and Equity, University of New South Wales, Kensington, NSW 2052, Australia; unyadav1@gmail.com; 3BRAC James P Grant School of Public Health, BRAC University, Dhaka 1213, Bangladesh; 4Department of Public Health, Daffodil International University, Dhaka 1207, Bangladesh; 5Global Research and Data Support, Innovations for Poverty Action, New Haven, CT 06510, USA; 6National Centre for Epidemiology and Population Health, Research School of Population Health, The Australian National University, Canberra, ACT 2600, Australia; 7School of Health Sciences, Western Sydney University, Sydney, NSW 2560, Australia; hudasoc2020@gmail.com; 8The School of Liberal Arts and Social Sciences, Independent University, Dhaka 1229, Bangladesh; 9Department of Sociology and Gerontology and Scripps Gerontology Center, Miami University, Oxford, OH 45056, USA; ghimirs2@miamioh.edu; 10Development Studies Discipline, Khulna University, Khulna 9208, Bangladesh; 11Institute of Nutrition and Food Science, University of Dhaka, Dhaka 1000, Bangladesh; sompa.infs@gmail.com; 12Department of Economics, University of Dhaka, Dhaka 1000, Bangladesh; rumanah14@yahoo.com; 13ARK Foundation, Gulshan, Dhaka 1212, Bangladesh; 14School of Health, Federation University Australia, Berwick, VIC 3350, Australia; ma.rahman@federation.edu.au; 15Department of Non-Communicable Diseases, Bangladesh University of Health Sciences (BUHS), Dhaka 1216, Bangladesh; 16Faculty of Public Health, Universitas Airlangga, Surabaya 60115, Indonesia

**Keywords:** Rohingya, tobacco use, smoking, smokeless tobacco, COVID-19

## Abstract

This study explored the perceived change in tobacco use during the COVID-19 pandemic and its associated factors among older adults residing in Rohingya refugee camps, also referred to as Forcibly Displaced Myanmar Nationals in Bangladesh. The study followed a cross-sectional design and was conducted in October 2020 among 416 older adults aged 60 years and above. A purposive sampling technique was applied to identify eligible participants, and face-to-face interviews were conducted using a pre-tested semi-structured questionnaire to collect the data. Participants were asked if they noted any change in their tobacco use patterns (smoking or smokeless tobacco) during the COVID-19 pandemic compared to pre-pandemic. Binary logistic regression models determined the factors associated with the perceived change in tobacco use. More than one in five participants (22.4%) were current tobacco users, of whom 40.8% reported a perceived increase in tobacco use during the COVID-19 pandemic. Adjusted analysis revealed that participants who were concerned about COVID-19 had significantly (*p* < 0.05) lower odds of perceived increase in tobacco use (aOR = 0.22, 95% CI: 0.06–0.73), while older adults who were overwhelmed by COVID-19 (aOR = 0.26, 95% CI: 0.06–1.18) and communicated less frequently with others during the pandemic than before (aOR = 0.19, 95% CI: 0.03–1.20) had marginally significantly (*p* < 0.1) lower odds of perceived increase in tobacco use during this pandemic. Relevant stakeholders, policymakers, and practitioners need to focus on strengthening awareness-raising initiatives as part of an emergency preparedness plan to control tobacco use during such a crisis period.

## 1. Introduction

Tobacco use is considered one of the most common causes of deaths and illnesses among people of all ages and is of particular concern among the older population [1,2]. The use of tobacco products is estimated to account for approximately 8% (or 4.9 million) of annual global deaths and 4.1% of disability-adjusted life years [3]. Research has documented that both smoking and using smokeless tobacco products (SLT) are the major risk factors of non-communicable diseases worldwide [2,4]. Among older adults, tobacco use has also been related to an increased risk of cognitive impairment, dementia, several sensory problems, loss of function, mobility, and independence [5,6]. Despite the well-established detrimental effects of tobacco use, we believe its use may have been increased during the pandemic. The pandemic has exacerbated serious mental health problems, including increased anxiety, depression, loneliness, and panic [7]. Imposed social isolation and prolonged lockdown during the pandemic and concomitant adverse psychological consequences can result in unhealthy habits, including increased tobacco use as a coping strategy [8,9]. The combination of biological and socioecological factors coupled with limited access to health care services during the pandemic may accelerate the level of depression, violence, suicide, and psychiatric illnesses [10], triggering increased tobacco use amid this pandemic.

The advent of the COVID-19 pandemic reminded us how interconnected the world is and that COVID-19 knows of no border and affects people on both sides of the border, including those at refuge [5]. COVID-19 poses an increased threat to refugees and displaced persons as they live in densely populated camps, making social distancing, hand washing, and personal hygiene difficult to maintain [11]. The Rohingya people, also known as Forcibly Displaced Myanmar Nationals (FDMNs), are Muslim minorities from the Rakhine state of Myanmar. Following attack by Myanmar militants, they started to flee in large numbers in August 2017 [12] and took shelter in the refugee camps in Cox’s Bazar, the south-eastern district of Bangladesh [13]. According to the latest report from the United Nations High Commissioner for Refugees, a total of 860,365 inhabitants are residing in the camp; 4% of them are older than 60 years [14]. Like many other refugee camps, population density is very high in Rohingya camps, ranging from 40,000 to 70,000 individuals per square kilometre [15], facilitating the rapid spread of coronavirus [16]. As of 3 January 2021, there were 367 confirmed COVID-19 cases and 10 deaths in camps [17].

Older adults are particularly subjected to behavioural risk factors, including tobacco consumption, due to many factors such as education, culture, and mental health conditions [18,19], predisposing them to increased chronic illnesses [20,21]. Older adults from the refugee background may be particularly vulnerable to increased mental health conditions during this pandemic [22] with a possible increase in tobacco use and chronic disease episodes. While a few recent studies [23,24] have documented adverse mental health issues among the Rohingya older adults, no research has reported their tobacco use patterns amid this pandemic. Therefore, the present research aimed to explore the perceived change in tobacco use and the factors associated with this change among older adults residing in the Rohingya refugee camps during this COVID-19 pandemic in Bangladesh.

## 2. Materials and Methods

### 2.1. Study Design and Participants

A cross-sectional survey was conducted in October 2020 among Rohingya older adults residing in refugee camps in Bangladesh’s Cox’s Bazar. The sample size of 460 was calculated with the following assumptions: (unknown) prevalence of COVID-19 = 50%, sampling error = 5%, confidence interval = 95% and non-response rate = 20%. Of the 460 approached eligible subjects, 416 Rohingya older adults aged 60 and above participated (response rate 91%). Of the total 34 Rohingya camps established in Cox’s Bazar district, Camp 08E (SSID CXB-210) was purposively selected because we had permission to access this camp only, which resembles other camp environments. We also adopted a purposive sampling technique to identify eligible participants from the selected camp in the absence of a suitable sampling frame (e.g., a list of the older adults) in Rohingya camps. Such purposive sampling is often chosen to conduct surveys among marginalised and hard-to-reach populations [25]. The data collectors continued visiting households, starting from one side of the camp and stopping once the desired sample size was achieved. One eligible participant was interviewed from each household. The oldest individual was interviewed in cases of more than one eligible participant in the selected household. In the absence of an eligible participant in the selected household, the data collectors moved to the next one. Participants aged ≥60 and having Rohingya FDMN status were included in the study. We excluded older individuals who were unable to communicate. The study sampling procedure is presented in Figure 1.

### 2.2. Data Collection Tools and Techniques

A pre-tested semi-structured questionnaire in the Bengali language was used to collect information through face-to-face interviews. Data were electronically recorded in Survey CTO mobile app (Dobility, Inc., Washington, DC, USA) (https://www.surveycto.com/, accessed on 13 November 2021) by two surveyors who were local residents of Cox’s Bazar, fluent in Rohingya dialects, and had previous experience in administering health surveys using electronic platforms. Prior to data collection, the data collectors were trained extensively through a half-day Zoom meeting. The data collectors were trained on data collection tools, techniques, and procedures for maintaining COVID-19 safe behaviors during the data collection.

The English version of the questionnaire was first translated to the Bengali language and then back-translated to English by two researchers to ensure the contents’ consistency. The Bengali version of the tool was piloted among a small sample (n = 10) of Rohingya older adults from the selected camp to refine the language in the final version. No changes were made following the pre-test in the final tool. Data collection was accomplished using the final tool through face-to-face interviews of the participants. Each interview took around half an hour.

### 2.3. Outcome Measure

The outcome variable of this study was the perceived change in tobacco use among current tobacco users (smoking any tobacco products or using any smokeless tobacco (SLT) within 30 days of the survey) [26] during the COVID-19 pandemic. We asked the participants if they had noted any changes in their tobacco use patterns during the COVID-19 pandemic compared to that of before the pandemic. Since no participants indicated decreasing their tobacco use during the COVID-19 pandemic, the outcome was treated as a binary variable where no change was coded as 0, and an increase in tobacco use was coded as 1.

### 2.4. Explanatory Variables

An extensive review of available studies guided the selection of explanatory variables [2,18,19,27]. Explanatory variables included age (categorised as 60–69 and ≥70 years), sex (male/female), marital status (married/widowed), family size (≤4 and more than 4), living arrangements (living with family/living alone), walking distance to the nearest health centre (<30 min/≥30 min), frequency of communication during COVID-19 (same as previous/less than previous), perceived difficulties obtaining food, medications and routine medical care during COVID-19 (yes/no), feeling of loneliness (hardly/sometimes to often), perception that older adults required additional care during COVID-19 (yes/no) and perception that they are at highest risk of COVID-19.

Self-reported information on pre-existing medical conditions, such as arthritis, hypertension, heart diseases, stroke, hypercholesterolemia, diabetes, chronic respiratory diseases, chronic kidney disease, and cancer, was also collected and recoded into a binary variable, pre-existing non-communicable chronic conditions (yes/no), indicating the absence of all of these conditions or presence of one or more of these conditions.

### 2.5. Statistical Analysis

The distribution of the variables was assessed through descriptive analysis. Given the categorical nature of variables, Chi-square tests were performed to compare differences in perceived increase in tobacco use by explanatory variables, with a 5% level of significance. We used binary logistic regression models to explore the factors associated with the perceived increase in tobacco use. The initial model was run with all potential covariates, listed in Table 1, and a final model was selected using the backward elimination based on the Akaike information criterion (AIC) (Table 2). The adjusted odds ratio (aOR) and associated 95% confidence interval (95% CI) are reported. All analyses were performed using the statistical software package Stata (Version 14.0).

## 3. Results

### 3.1. Characteristics of the Participants

A total of 416 older adults participated in this study, of whom 93 participants (22.4%) used tobacco, i.e., smokers (8.2%), SLT users (15.6%), and dual users (1.4%). Of the 93 current tobacco users, 79.6% were between ages 60–69 years, 69.9% were male, 96.8% were married, 64.5% had a family size of 4 or more-person, 94.6% were living with family, and 80.7% resided in less than 30 min walking distance from the nearest health centre. Details are shown in Table 1.

### 3.2. Changes in Tobacco Use during COVID-19

None of the participants reported initiating tobacco use or decreased use (among current smokers) during this pandemic. However, 40.9% of the current tobacco users reported increased use of tobacco during COVID-19. More specifically, the increase in smoking and use of SLT products were reported by 32.4% of the current smoker and 47.7% of the current SLT users, respectively. Statistically, there were no significant differences by sex, but participants reporting difficulty accessing medicine and routine medical care during the COVID-19 pandemic were more likely to report increased tobacco use (*p* < 0.05). Meanwhile, perceived increase in tobacco use was significantly lower among the participants who were concerned about and overwhelmed by COVID-19 (*p* < 0.001). Please see details in Table 1.

### 3.3. Factors Associated with Changed Tobacco Use during COVID-19

In the adjusted model, participants who were concerned about COVID-19 had significantly lower odds (*p* < 0.05) of perceived increased tobacco use (aOR = 0.22, 95% CI: 0.06–0.73), while the older adults who were overwhelmed by COVID-19 (aOR = 0.26, 95% CI: 0.06–1.18), and who communicated less frequently with others during the pandemic than before (aOR = 0.19, 95% CI: 0.03–1.20) may also have lower odds of perceived increase in tobacco use during this pandemic; however, confidence intervals showed the possibility of no difference. We also found that, although not statistically significant, perceived tobacco use was increased among the current tobacco users who had difficulties accessing medication during the COVID-19 pre-pandemic (aOR = 5.39, 95% CI: 0.66–44.24), and who had perceived loneliness (aOR = 3.98, 95% CI: 0.76–20.93). Details are presented in Table 2.

## 4. Discussion

This study examined the change in tobacco consumption patterns and associated factors during the COVID-19 among older adults residing in the Rohingya refugee camps in Cox’s Bazar district of Bangladesh. Our study found that around 41% of the current tobacco users among the older Rohingya population reported increased tobacco use during the COVID-19 pandemic. We did not find any single study on tobacco use during the pandemic among Rohingya older adults in Bangladesh and other countries to compare this finding with. However, a recent study on Bangladeshi older adults indicated around a 16% increase in tobacco consumption during the COVID-19 pandemic [27], which is lower than the current study’s findings. Furthermore, a Somalian study reported 3% of tobacco products or cigarette use among internally displaced persons (IDP) camps, which is very low compared with the current study. One probable reason for such a low consumption of tobacco products is that women constituted a majority of the sample (86%) of the Somalia study [28], who smoke less than men [29,30]. Contrarily, 22% of tobacco consumption was found among Syrian refugees in Turkey [31]. The pre-migration factors (e.g., displacement from home, limited social networks with relatives and friends, and prior poignant experience of torture and violence) deteriorated the psychological well-being of the Rohingya population [15]. Such pre-migration factors may contribute to increasing refugees’ risk of tobacco smoking because of the perception that tobacco consumption can reduce psychological distress [32,33]. Furthermore, the post-migration risk factors of Rohingya people (e.g., poor living conditions in camps, abuse, limited income opportunities, and perceived discrimination), coronavirus-related travel restrictions, and lockdowns further intensify their mental health problems [15], such as post-traumatic stress disorder, anxiety, hopelessness, and suicidal thoughts [34]. Their aggravated mental health conditions may lead to adopting harmful behaviours, including tobacco smoking [35]. Previous studies also reported similar issues, suggesting that perceived ethnic discrimination [36] and post-migration factors [32,37] may likely increase smoking rates among refugees, immigrants, and ethnic minority communities. Our findings suggest the necessity of developing tobacco cessation interventions in the context of Rohingya people living in camps in Cox’s Bazar, Bangladesh.

One potential reason for increased tobacco consumption among older adults from the Rohingya refugee camps is their inadequate and inaccurate knowledge about the fact that tobacco consumption causes harmful health effects and exacerbates comorbid conditions (e.g., stroke, heart disease, and lung cancer) [38]. This is similar to existing literature which suggests that smokers have limited accurate knowledge compared to non-smokers [39,40,41]. Such limited knowledge may lead smokers to spend extra money on cigarette smoking, further constraining their access to medications [28]. Furthermore, current older cigarette smokers may perceive the limited benefits of smoking cessation [42], prompting them not to cease smoking and continue it [43]. Smoking may increase individuals’ vulnerability to smoking-induced diseases (e.g., lung cancer, chronic bronchitis, coronary heart disease, etc.) [44,45] and COVID-19 infection by damaging the existing immune system and reducing its ability to respond to disease [46,47]. These health complications require multiple medications for current smokers whose healthy life expectancy may be impacted [48]. While coronavirus-related lockdown, inadequate transport, financial constraints, limited availability of medicines, and unavailability of medicines may limit their choices about accessing multiple medications, the availability of cheap tobacco products (e.g., bidi) may increase their tobacco consumption [15,27,49]. Therefore, relevant stakeholders, policymakers, health promotion educators, and program implementers should undertake campaigns for creating awareness about limiting tobacco consumption and offer cessation advice and they should undertake initiatives for facilitating access for Rohingya older adults to medicines during the pandemic.

Our study also suggests that older adults from the Rohingya refugee camps who were concerned about and were overwhelmed by COVID-19 were less likely to have increased tobacco use. The potential reason is that those concerned about the COVID-19 may spend more time with non-smokers (family members) during the lockdown, helping them to avoid tobacco products [50]. Furthermore, their tendency to limit smoking may be affected by the belief that coronavirus is more prevalent among smokers than non-smokers [50,51]. The older adults from the Rohingya refugee camps who were concerned about COVID-19 might also acknowledge that the COVID-19 has already disrupted them, reduced their income, overwhelmed their lives [15], increased their risks of COVID-19 [52], and worsened comorbid conditions [53]. Moreover, tobacco users concerned about COVID-19 might also be considerate about the lockdown and restrictions posed due to COVID-19 and restrain themselves from going outside to buy tobacco products.

The current study revealed that older adults from the Rohingya refugee camps having less frequent communication with others (e.g., family members, relatives, friends, etc.) during the pandemic than before had lower odds of increased tobacco use. Our finding contradicts two commonly held opinions that strong interpersonal communication can contribute to smoking cessation [54], and decreased communications can lead to increased tobacco consumption [27]. We did not come across any study on the refugee population globally to compare our findings. However, our study’s finding confirms prior research on adult tobacco users in the United States, suggesting that adults with little interpersonal communication tended to consume less tobacco products [55]. The potential reasons for a lower chance of tobacco use among older adults from the Rohingya refugee camps were that coronavirus-related lockdown measures might limit Rohingyas’ movement and frequency of interpersonal communication with moderate or heavy smokers [56]. Moreover, frequent interactions with non-smokers and children at the family level and support from family members during lockdown may motivate them to cease tobacco use [50]. Additionally, some Rohingya older adults may take coronavirus-related lockdown measures as an opportunity to quit smoking [56].

### 4.1. Policy Implications

The current study’s findings are important because they can help identify the entry points for designing interventions for minimizing tobacco consumption among Rohingya older adults. The study findings highlight the necessity for changing the behaviors of older adults from the Rohingya refugee camps during this COVID-19 pandemic. The findings point to the importance of focusing on strengthening awareness-raising initiatives among this vulnerable population as part of an emergency preparedness plan, specifically emphasizing controlled tobacco use among the current tobacco users during this pandemic. The Office of the Refugee Relief and Repatriation Commissioner under the Government of Bangladesh and international organizations (e.g., the United Nations High Commissioner for Refugees, International Organization for Migration, and International Committee of the Red Cross, etc.) can play a vital role in minimizing Rohingyas’ increased tobacco consumption during the pandemic.

### 4.2. Strength and Limitations

Our study has several strengths. First, to our knowledge, this study is the first of its kind to explicitly examine the perceived change in tobacco use during COVID-19 and associated factors among Rohingya older adults in Bangladesh. This study adds to the limited international literature exploring tobacco consumption patterns and associated factors among the older adults from Rohingya refugee camps during the pandemic. Second, our study area and population are unique because the Rohingya refugee camps are the biggest globally, and the Rohingya population include one of the most persecuted minorities and represent other refugee populations living in similar environments across the world [57]. Despite these strengths, our study findings should be interpreted in the context of several limitations. First, data on tobacco use and associated factors were all self-reported, which are often subject to non-disclosure. Second, our study used purposive sampling for selecting study sites and participants from a single camp only, which could introduce selection bias and consequently limit the generalizability of the findings to the entire population of Rohingya older adults in the camps.

Third, our study was cross-sectional in nature. Therefore, causality cannot be established. Fourth, our study is based on quantitative analysis. We did not explore the qualitative aspects of tobacco use patterns and their associated factors among the older adults from the Rohingya refugee camps during the COVID-19. These limitations highlight the need for further studies with a mixed-methods approach, including a qualitative study exploring tobacco consumption patterns and their associated factors among the older adults from the Rohingya refugee camps during the COVID-19 pandemic. This will provide a better understanding of tobacco consumption patterns and their associated factors among older adults from Rohingya refugee camps during the COVID-19 pandemic in Bangladesh.

## 5. Conclusions

This paper highlighted that more than 40% of older adults from Rohingya refugee camps reported increased tobacco use amid this pandemic. This was significantly higher among those less concerned about the COVID-19 and those who constantly communicated with friends during the pandemic. The findings of the study suggest strengthening awareness and raising initiatives as part of an emergency preparedness plan to reduce tobacco use in refugee camps, particularly among this vulnerable group.

## Figures and Tables

**Figure 1 ijerph-18-12349-f001:**
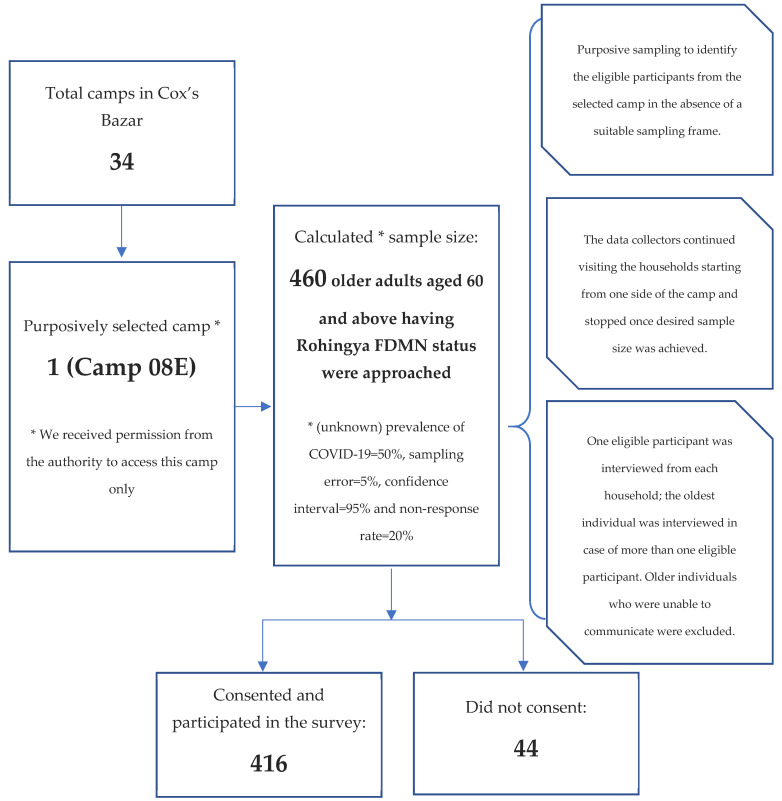
Sample selection and study participants.

**Table 1 ijerph-18-12349-t001:** Changes in frequency of tobacco consumption during COVID-19 (N = 416).

Characteristics		Overall (N = 416)	Tobacco User (n = 93)	Change in Tobacco Use (n = 93)		
				No Change	Increased	
		n (%)	n (%)	n (%)	n (%)	*p*
Overall				55(59.1)	38(40.9)	
Age (year, %)						
	60–69	308 (74.0)	74 (79.6)	42(56.8)	32 (43.2)	0.356
	≥70	108 (26.0)	19 (20.4)	13 (68.4)	6 (31.6)	
Sex						
	Male	251 (60.3)	65 (69.9)	41 (63.1)	24 (36.9)	0.239
	Female	165 (39.7)	28 (30.1)	14 (50.0)	14 (50.0)	
Marital status						
	Married	389 (93.5)	90 (96.8)	53 (58.9)	37 (41.1)	0.787
	Widowed	27 (6.5)	3 (3.2)	2 (66.7)	1 (33.3)	
Family size						
	≤4	167 (40.1)	33 (35.5)	21 (63.6)	12 (36.4)	0.513
	>4	249 (59.9)	60 (64.5)	34 (56.7)	26 (43.3)	
Living arrangement						
	Living with family	362 (87.0)	88 (94.6)	52 (59.1)	36 (40.9)	0.968
	Living alone	54 (13.0)	5 (5.4)	3 60.0)	2 (40.0)	
Walking distance to the nearest health centre						
	<30 min	330 (79.3)	75 (80.7)	46 (61.3)	29 (38.7)	0.380
	≥30 min	86 (20.7)	18 (19.4)	9 (50.0)	9 (50.0)	
Concerned about COVID-19						
	Hardly	256 (61.5)	51 (54.8)	24 (47.1)	27 (52.9)	0.009
	Sometimes/often	160 (38.5)	42 (45.2)	31 (73.8)	11 (26.2)	
Overwhelmed by COVID-19						
	Hardly	167 (49.1)	55 (64.0)	28 (50.9)	27 (49.1)	0.002
	Sometimes/often	173 (50.9)	31 (36.1)	26 (83.9)	5 (16.1)	
Frequency of communication during COVID-19						
	Same as previous	170 (40.9)	62 (66.7)	37 (59.7)	25 (40.3)	0.881
	Less than previous	246 (59.1)	31 (33.3)	18 (58.1)	13 (41.9)	
Difficulty in getting food during COVID-19						
	No	264 (67.7)	67 (76.1)	41 (61.2)	26 (38.8)	0.953
	Yes	126 (32.3)	21 (23.9)	13 (61.9)	8 (38.1)	
Difficulty in getting medicine during COVID-19						
	No	277 (69.8)	73 (79.4)	47 (64.4)	26 (35.6)	0.030
	Yes	120 (30.2)	19 (20.7)	7 (36.8)	12 (63.2)	
Difficulty receiving routine medical care during COVID-19						
	No	275 (70.0)	74 (83.2)	48 (64.9)	26 (35.1)	0.006
	Yes	118 (30.0)	15 (16.9)	4 (26.7)	11 (73.3)	
Perceived that older adults are at highest risk of COVID-19						
	No	151 (36.3)	59 (63.4)	35 (59.3)	24 (40.7)	0.962
	Yes	265 (63.7)	34 (36.6)	20 (58.8)	14 (41.2)	
Feeling of loneliness						
	Hardly	332 (79.8)	76 (81.7)	50 (65.8)	26 (34.2)	0.006
	Sometimes/often	84 (20.2)	17 (18.3)	5 (29.4)	12 (70.6)	
Perceived that older adults required additional care during COVID-19						
	No	314 (75.5)	82 (88.2)	53 (64.6)	29 (35.4)	0.003
	Yes	102 (24.5)	11 (11.8)	2 (18.2)	9 (81.8)	
Pre-existing non-communicable chronic conditions						
	No	295 (70.9)	65 (69.9)	40 (61.5)	25 (38.5)	0.473
	Yes	121 (29.1)	28 (30.1)	15 (53.6)	13 (46.4)	

**Table 2 ijerph-18-12349-t002:** Factors associated with increased tobacco use during COVID-19 (N = 93).

Characteristics		AOR	95% CI	*p*
Family size				
	≤4	Ref		
	>4	2.49	0.74–8.30	0.138
Concerned about COVID-19				
	Hardly	Ref		
	Sometimes/often	0.22	0.06–0.73	0.014
Overwhelmed by COVID-19				
	Hardly	Ref		
	Sometimes/often	0.26	0.06–1.18	0.081
Frequency of communication during COVID-19				
	Same as previous	Ref		
	Less than previous	0.19	0.03–1.19	0.077
Difficulty in getting food during COVID-19				
	No	Ref		
	Yes	0.21	0.03–1.51	0.121
Difficulty in accessing medicine during COVID-19				
	No	Ref		
	Yes	5.40	0.66–44.24	0.116
Feeling of loneliness				
	Hardly	Ref		
	Sometimes/often	3.98	0.76–20.93	0.103
Perceived that they required additional care during COVID-19				
	No	Ref		
	Yes	5.43	0.43–68.30	0.190

## Data Availability

The data are available upon reasonable request from the corresponding author.

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
