# Peer review of "Perceived Change in Tobacco Use and Its Associated Factors among Older Adults Residing in Rohingya Refugee Camps during the COVID-19 Pandemic in Bangladesh"

_ijerph, 2021, doi:10.3390/ijerph182312349_

Round 1

Reviewer 1 Report

The article is particularly interesting given the high-stress population due to known living conditions. The sample is unfortunately not very large, the active part taken into consideration is only 93 individuals, despite the largest cohort in total; despite this, several strengths are noted, appropriately highlighted in the study. It would be interesting to comment in more depth on the link between additional diseases, comorbidities, and smoking, with a view to identifying lower life expectancy or hope, which could be in line with the sometimes unfortunately marked habit of some segments of the population not to adhere perfectly to the use of personal protective equipment such as masks for covid-19, or tobacco intake.
Overall the article is well written, with some adjustments as suggested I think it could be more complete

Reviewer 2 Report

The research topic is interesting but several major changes are needed:

  1. Please clearly define that this study is limited to the "older 
    adults residing in refugee camps in Bangladesh" - this information should be included in the title as well as other parts. This is crucial information due to the fact, that this study is limited to the selected group of adults (refugees) and the results can not be generalized to the whole population
  2. Please consider adding study flow (including sampling methods etc.)
  3. Please justify the importance of this study for international readers. The current version of the manuscript is limited to some local data, and the current version of the manuscript does not match with the IJERPH objectives.
  4. Please justify the study sample - why this study was carried out among older adults (refugees) in one location in Bangladesh. The journal is international so this should be clearly presented.
  5. Potential practical implications of this study should be clearly defined
  6. Please clearly explain the novelty and importance of this study 

Round 2

Reviewer 1 Report

Dear Editor,
I really appreciate the opportunity to review the manuscript  entitled:
"Perceived change in tobacco use and its associated factors among Rohingya older adults residing in Rohingya refugee camps during the COVID-19 pandemic in Bangladesh"

The paper is very interesting and well-written, methodologically unexceptionable, and the new implementations provide a valid contribution to the work. Every requested correction has been done, and the manuscript is now suitable for publication

Reviewer 2 Report

The manuscript was significantly revised. The Authors addressed all the comments.